# Recent Advances in the Management Strategies for Buruli Ulcers

**DOI:** 10.3390/pathogens12091088

**Published:** 2023-08-27

**Authors:** Gabriela Loredana Popa, Alexandru Andrei Muntean, Mircea Ioan Popa

**Affiliations:** 1Department of Microbiology, Faculty of Dental Medicine, “Carol Davila” University of Medicine and Pharmacy, 050474 Bucharest, Romania; 2Colentina Clinical Hospital, 020125 Bucharest, Romania; 3Department of Microbiology II, Faculty of Medicine, “Carol Davila” University of Medicine and Pharmacy, 050474 Bucharest, Romania; alexandru.muntean@umfcd.ro (A.A.M.); mircea.ioan.popa@umfcd.ro (M.I.P.); 4“Cantacuzino” National Military Medical Institute for Research and Development, 050096 Bucharest, Romania

**Keywords:** Buruli ulcer, *Mycobacterium ulcerans*, treatment

## Abstract

Buruli ulcer (BU) is a bacterial skin infection that is caused by *Mycobacterium ulcerans* and mainly affects people who reside in the rural areas of Africa and in suburban and beach resort communities in Australia. The infection typically begins as a painless papule or nodule that gradually develops into a large ulcer that can cause substantial impairment, damaging soft tissues and even bones. Early detection and immediate treatment are crucial to preventing further tissue damage and any potential complications, although it is worth noting that access to proper therapeutic resources can be limited in certain areas. The most commonly used antibiotics for treating BU are rifampicin, streptomycin, and clarithromycin; efforts have recently been made to introduce new treatments that increase the effectiveness and adherence to therapy. This article presents the latest research and management strategies regarding BU, providing an updated and intriguing perspective on this topic.

## 1. Introduction

*Mycobacterium ulcerans* (*M. ulcerans*) is a slow-growing mycobacterium and the causative agent of Buruli ulcer (BU), one of the most neglected tropical diseases [1]. The bacteria can be cultured in vitro at 32 °C using standard media for the mycobacterial culture. Whole-genome sequencing analyses revealed that *M. ulcerans* arose from ubiquitous fast-growing *M. marinum*, a nontuberculous bacteria that cause skin infections through the acquisition of a virulence plasmid (pMUM), which contains the genes responsible for the enzymes necessary for the production of macrolide toxins called mycolactones. The evolution of *M. ulcerans* involves reductive processes and the formation of pseudogenes, possibly to adapt to a more stable ecological niche, facilitated via the proliferation of specific insertion sequence (IS) elements in its genome, such as 213 copies of IS2404 and 91 copies of IS2606 [2,3].

The strains isolated from the specific regions exhibit notable similarity, but variations between geographical areas have been observed, particularly regarding the type of mycolactone synthesis. These differences may indicate regional variations in the clinical presentation of *M. ulcerans* infection. In the early stages of *M. ulcerans* infection, a significant number of extracellular bacilli and considerable necrosis are evident. However, the inflammatory response is reduced, and no granulomas are detected. In the later stages, when the healing process begins, after effective antibiotic treatment, the bacilli are present in small numbers, and multiple granulomas may be observed [4].

BU is the third most prevalent human mycobacteriosis [5,6]. BU infections have been reported in 34 countries, the majority of which are in the West African region and in Southeastern Australia, where it affects suburban and beach resort (coastal) communities [7]. Albert Cook, an English physician, was the first to observe the disease during the late nineteenth century, (as found in his notes in a hospital library in Kampala, Uganda) but he never reported the disease. It typically presents with cutaneous lesions that later evolve into painless ulcers, causing significant tissue damage. One of the main virulence factors of *M. ulcerans*, involved in the pathogenesis of BU, is mycolactone—an exotoxin that induces apoptosis and blocks cell cycle progression. Severe lesions occur in about a third of cases, leading to disability and social stigma [8,9,10]. Adamba et al. showed that over 62% of patients with BU feel stigmatized [11]. On the other hand, other studies indicate that communities still offer empathy to patients with BU [12,13]. Owusu et al. have highlighted that the households experience substantial socioeconomic burden due to BU, which is evident through three primary dimensions: health-related, financial, and socio-psychological impacts [14]. BU has the potential to adversely affect school attendance and the educational progress of every child. Due to delayed medical treatment, in severe cases, the illness can result in disabilities such as amputation of a leg, leading to school drop-out [15,16].

World Health Organization (WHO) classified BU into three categories (Table 1). 

The gold standard for BU diagnosis remains microbiological confirmation, which consists of quantitative PCR targeting the *M. ulcerans* insertion sequence IS2404 performed on cutaneous samples. However, the PCR confirmation rate in Africa was very low, at only 30% in 2018. In this regard, 11 laboratories established the BU-LABNET network in 2019. The initial step in creating the network was to standardize the laboratory procedures and then distribute specific reagents to each facility. Once this system was launched, implementing the testing and follow-up procedures was straightforward, and the laboratories were able to conduct their initial quality control with an increased success rate. Since its beginning in 2019, two additional laboratories have become part of BU-LABNET: the West African Center for Cell Biology and Infectious Pathogens (WACCBIP) in Ghana (an academic laboratory) and St Joseph’s Hospital Adazi Nnukwu in Nigeria (a private institution) [18]. Despite the efficient use of antibiotics in the management of BU patients, several therapeutic problems remain. In this narrative review, we summarize the recent advances made towards overcoming these challenges in the management of BU.

## 2. The Transmission Route—An Enigma Still to Be Solved

Many questions remain unanswered regarding the infection’s transmission route. To date, human-to-human transmission has not been established in the case of *M. ulcerans*. However, there are some reports attesting unusual human-to-human transmission after an accidental injection or bite [19,20,21]. Non-human cases of BU were first identified in Australia, with the presence of the microorganism confirmed in possums [22]. Hence, mammals native to Southeastern Australia, like possums, are carriers of *M. ulcerans*, the same bacteria that cause human infection [23]. Genotyping has confirmed that these mammals are part of the transmission network for the disease. However, information on mammalian hosts in West Africa is limited. Only human cases have been detected in Africa, but researchers draw attention to the importance of studying domestic and wild animals in endemic regions in order to understand whether or not they can represent a transmission source [22]. The reported cases of BU in wild and domesticated animals such as koalas, possums, and alpaca in Australia validate the ability of *M. ulcerans* to affect certain animal species [24].

Mosquitoes are thought to be the primary vector of *M. ulcerans* transmission in Southeastern Australia, although other biting aquatic insects may be involved in African nations. Human disturbances to the African environment might generate favorable circumstances for *M. ulcerans*, and outbreaks of BU are occasionally preceded by such disruptions [10].

Drancourt et al. investigated why the prevalence of BU has decreased in Africa in recent years and concluded that temperature differences might impact this phenomenon. It is universally assumed that stagnant water environments are reservoirs for *M. ulcerans* in Africa. Since global warming has led to the drying up of some stagnant water sources, it has decreased people’s exposure to infected water sources [25]. Temperature fluctuations are known to cause epidemic outbreaks of infectious illnesses, particularly when infections are spread via vectors. Since *M. ulcerans* may be spread by insects, the temperature variations recorded in recent years may influence the population, distribution, and habitat of insects, with consequences on the incidence of BU [25]. However, the most important factor that influences the incidence of BU is the failure to detect cases. Ahorlu et al. showed that more cases can be found with active case detection [26]. 

## 3. Diagnostic Approaches and Challenges

For the confirmation of a Buruli ulcer diagnosis, four main methods are available: microscopic examination for the detection of acid-fast bacilli, cultivation on specific culture media, PCR targeting specific *M. ulcerans* genes, and histopathological examination [27].

Microscopy is a simple, rapid, and cost-effective method. The examined samples include swabs from ulcerative lesions, tissue from biopsies or surgical excision, and fine-needle aspirates. Typically, Ziehl–Neelsen staining is used to visualize acid-fast bacilli, and quantification methods similar to those used in tuberculosis diagnosis are employed. Other possible staining methods are Kinyoun and auramine–rhodamine. The efficiency of the method depends on the skills of the microscopist and the performance of the microscopy equipment. Microscopic examination has reduced sensitivity and specificity (30–40%), but in the context of a patient with classical Buruli ulcer lesions from an area where many cases are reported, it is highly suggestive. It should be noted that *M. marinum*, a non-tuberculous mycobacterium, can also produce skin lesions [9,27,28]. 

In terms of specificity, the culture has a high specificity, but the results take a long time to obtain. It has reduced sensitivity, 35–50%, and false-negative results can occur. The results are dependent on the site of tissue sample collection (bacilli are present in deep tissues), the decontamination method, and the culture media used [27,29,30].

PCR is the most sensitive method, detecting between 54 and 84% of cases, but standardized protocols are not available, and it should be noted that it may detect nonviable bacteria. The most widely used PCR methods are conventional single-step gel-based PCR and real-time PCR targeting the insertion element IS2404.Other *M. ulcerans*-specific sequences are IS2606 and the ketoreductase-B domain of the mycolactone polyketide synthase genes. False negative test results may occur due to various factors, such as a limited amount of *M. ulcerans* DNA in the lesion samples, suboptimal DNA extraction efficiency, reduced PCR sensitivity, or the presence of PCR inhibitors. Unfortunately, in regions lacking medical facilities, the diagnosis of Buruli ulcer is often based on clinical appearance. PCR and culturing *M. ulcerans* may not be readily available in resource-limited settings [9,27,28].

Histopathological examination has the advantage of establishing the diagnosis of Buruli ulcer and also helps in the differential diagnosis (for example, squamous cell carcinoma or fungal infection). Well-defined areas of contiguous coagulation necrosis are observed in the dermis and subcutaneous tissue, but sometimes the lesions can be very deep and reach the level of the fascia. One of the most significant disadvantages of histopathological examination is that it requires experienced laboratory personnel. False-negative results may occur if the tissue sample is superficial [29,31]. 

Alternative methods for diagnosis have also been investigated. Fluorescent thin-layer chromatography for mycolactone shows a very good, over 70%, sensitivity and specificity in the mouse studies [32]. It is true that it is not easily conducted in all labs, and further studies are necessary. Regarding serological diagnosis, the results are not clear. It is worth mentioning that antibodies against *M. ulcerans* have been identified in healthy contacts [27].

Enhancing awareness among healthcare professionals, ensuring the availability of adequate healthcare resources and diagnostic equipment, and conducting research to create more sensitive and easily accessible diagnostic techniques for Buruli ulcer are crucial.

## 4. Current Buruli Ulcer Treatment and the Main Challenges to Completion

Given the failures of the early research on antibiotic therapies for BU up to 2004, surgery was the treatment of choice, with a mutilating impact in areas such as the face or genitals. Later, the antibiotic treatment based on the combination of streptomycin and rifampicin (SR8 regimen) was introduced [33]. In the era of antibiotics, surgery no longer has a firmly established place [34]. It has been observed that following antibiotic therapy, the lesions might worsen, which is frequently taken as a treatment failure and leads to the decision to undergo surgery [35]. The most frequent complication of antibiotic treatment is the paradoxical reaction that involves the worsening of existing lesions, pain, and the development of new lesions. It can occur during or after treatment. It is presumed that the paradoxical reaction is the result of the immunological response to residual *M. ulcerans* antigens that can persist for a long period of time even after a successful outcome [36]. Nienhuis et al. suggested that “the proinflammatory response we describe as paradoxical response coincides with the wash-out of mycolactone from the lesion” [37]. The lesions that develop more than one year after completion of antibiotic therapy may be linked to new infection foci that are eradicated by the immune responses triggered by the effective treatment of the initial lesion [38].

A recent study showed that patients who develop a paradoxical reaction have an increased bacterial load at the site of the lesions and a high rate of positive *M. ulcerans* culture. Additionally, these patients experienced delayed healing of the lesions [39]. It appears that approximately 20% of patients undergoing antibiotic treatment for BU develop a paradoxical reaction, with the most common occurrence observed at 6–10 weeks after initiating therapy. There are no markers to predict the onset of the paradoxical reaction. As a result, a group of researchers showed that these patients have elevated serum levels of IL-6 and tumor necrosis factor alpha (TNF-α) on days 30, 60, and 90 after starting antibiotic treatment. Thus, maintaining elevated levels of IL-6 and TNF-α during treatment should be a warning signal for physicians regarding the possibility of the paradoxical reaction occurring [40].

WHO recommends to decide whether surgery is needed 4 weeks after starting the antibiotic treatment [17,35]. A retrospective study conducted by Wadagni et al. in Ghana and Benin included 1193 patients with BU and showed significant differences in the choice to perform surgical interventions. The probability of undergoing surgery was significantly higher for patients treated at one of the medical centers in Benin when compared to the clinic in Ghana with the lowest rate of surgical procedures. Even after adjusting for illness severity, age, gender, and other factors, the discrepancies remained significant. Therefore, further studies are needed to establish the optimal timing for surgery and the characteristics of eligible patients [35]. O’Brien et al. draws attention to the research on patients with BU in Australia, which shows the re-emergence of surgery as an important treatment option. Moreover, in the case of small-sized lesions, surgery alone can be a treatment modality [41]. Starting in 2017, BU can be treated orally with clarithromycin and rifampicin for 8 weeks (CR8 regimen), according to the latest WHO recommendations [42]. A systematic review examined the efficacy of rifampicin and streptomycin-based therapy for a period of 8–48 weeks depending on the disease severity (the mean duration—8 weeks) and found a 50% cure rate; when surgery was associated, the hospitalization period decreased by 44.2% [43]. Phillips et al. evaluated the CR8 and SR8 regimens and discovered that the prevalence of adverse events was 7% versus 13% in those who received oral therapy. The authors pointed out that CR8 led to similar results to SR8 in the case of early and small lesions. Injectable streptomycin therapy has the drawback of being unpleasant and linked with ototoxicity—streptomycin led to severe ototoxicity in one patient (1%) in this study [44]. Klis et al., using audiometry rather than patient reports, revealed that the extended usage of streptomycin among adults is associated with notable hearing impairment. Both adults and children experienced temporary nephrotoxicity. Therefore, administering streptomycin requires careful consideration, especially in patients aged 16 and above, as well as individuals with existing risks of renal dysfunction or hearing loss [45]. The study conducted by O’Brien et al. suggests that a 6-week treatment duration instead of 8 weeks as currently recommended is effective for patients with small-sized lesions, offering the advantages of lower toxicity associated with the therapy and reduced costs [46]. In Australia, the combination of rifampicin and moxifloxacin is sometimes used. Although the results obtained so far are satisfactory, there have been no randomized trials to support this [47].

A group of researchers from Australia evaluated in a prospective study the effectiveness of rifampicin in combination with ciprofloxacin, clarithromycin, or moxifloxacin (follow-up period—one year) and observed a cure rate of over 70% [48]. The combination of rifampicin, levofloxacin, and clarithromycin is used in Japan [9]. Microbiological studies on patients with BU revealed that there is additional colonization with *Staphylococcus* spp., *Bacillus* spp., and *Pseudomonas* spp., which can delay recovery [49]. Since the lesions are painless, patients frequently seek medical attention in later stages, when antibiotic therapy is less effective. Since the risk of sequelae associated with BU is high, the treatment should be started promptly and followed as prescribed. However, the data on treatment completion in Africa is scarce, and the results are contradictory. On the one hand, a study from Ghana carried out over a period of 5 years showed that the treatment was fully completed in less than 50% of cases (46%). On the other hand, another study identified a treatment completion rate of 90% [50,51,52]. Etuaful et al. has pointed out that many patients are culture negative after less than 8 weeks of treatment [53].

Research by Collinson et al. has revealed that adherence to treatment has increased since oral antibiotics have been prescribed as the first line of treatment. They have observed more serious forms in the cases of patients who live further away from medical facilities, which was to be expected. The study, which took place between 2006 and 2018, covered four clinics in Ghana, and the treatment completion rate was 84.4%, with a greater success rate reported in patients who followed the CR8 regimen compared to those who followed the SR8 regimen [50]. Klis et al. completed an equivalent study in the same region of Africa from 2008 to 2012, where all patients were treated with streptomycin and rifampicin, with a 46% success rate of treatment completion. The female gender, a smaller lesion size, and a greater travel time were the main factors associated with treatment interruption [51].

Adherence to therapy is influenced by various factors. A recent study focused on analyzing BU awareness among patients in a Ghanaian hospital and found concerning results with serious implications for the population’s control of this disease. The study consisted of 400 participants who completed a questionnaire, including patients with BU as well as individuals who were not suffering from this illness. Results showed that the participants had very little knowledge about BU. Many respondents believed that witches, enemies, and disrespecting the gods might cause BU to occur; nevertheless, more than half of the respondents stated that they were unaware of any risk factors for the emergence of BU. Many survey participants (30%) considered that an increased appetite was one of the symptoms of BU. Other symptoms identified by the participants included swelling of the skin and being overweight [54]. Ahorlu et al. emphasized the importance of implementing an active community-based surveillance-response system for early diagnosis of BU and proper treatment. The study was carried out in Africa, with subdistrict disease control officials, selected healthcare professionals, and trained community-based volunteers taking part. Each patient was examined 11 times over the course of a year. Out of 75 skin lesions, 12 were identified as BU using PCR testing. The survey also found that, towards the end of the trial, the community understanding of BU had grown. The implementation of such surveillance response systems can be very beneficial for BU control in Africa [26].

Involving the public is especially crucial for infection control. Communicating with members of the affected communities and maintaining a permanent link between them and the health personnel can significantly improve the control of neglected tropical diseases. It is important that the health personnel respect the community’s culture, but also address the cultural beliefs and practices that interfere with the early diagnosis and treatment of BU. Gaining the trust of community leaders is essential, and traditional healers should also be involved [55].

We should keep in mind the following paragraph that can be read on the official site of WHO: “The objective of Buruli ulcer control is to minimize the suffering, disabilities and socioeconomic burden. Early detection and antibiotic treatment are the cornerstones of the control strategy. In many countries, community health workers play a critical role in case detection” [47].

## 5. New Therapeutic Strategies for Buruli Ulcer

In the last years, efforts have been made to develop new molecules for the treatment of BU (Table 2). It has been revealed that *M. ulcerans* is susceptible to Q203 (telacebec), a compound that acts on respiratory cytochrome bc1:aa3. This agent is a candidate for the treatment of tuberculosis; cytochrome bc1:aa3 has been shown to be the primary terminal oxidase in *M. tuberculosis*. Yet, *M. tuberculosis* exhibits an alternate *bd*-type terminal oxidase, which decreases the bactericidal and sterilizing effects of Q203 against this bacterium. Conversely, research on *M. ulcerans* strains recovered from BU patients in Africa and Australia revealed that, due to a mutation in the genes encoding the *bd* oxidase, these strains lacked an alternate terminal oxidase, rendering these predominant *M. ulcerans* strains highly vulnerable to Q203. This indicates that Q203 may be a helpful antibacterial drug in this scenario [56]. It has been shown that a single dose of Q203 effectively eliminates *M. ulcerans* in a mouse model of BU, with no recurrence observed for up to 19 weeks after treatment. These findings strongly suggest that Q203 holds promise for single-dose or other very short therapeutic approaches for BU. However, in cases of highly immunocompromised individuals, it may be necessary to consider higher doses, longer durations, or combining Q203 with other therapies [57,58]. 

Chauffour et al. have verified the efficacy of a new group of antibiotics against *M. ulcerans* using a BU mouse model. They proposed that tedizolid, selamectin, ivermectin, and benzothiazinone PBTZ169 had no bactericidal effect. In contrast, telacebec had a bactericidal effect. Therefore, they have proposed a treatment scheme with telacebec in combination with rifapentine or bedaquiline, two times a week for 8 weeks, which led to the sterilization of mouse footpads and prevented relapses over a period of 20 weeks [59]. Another recent study on a mouse model has shown that telacebec in combination with rifampin for a period of 2 weeks is associated with a relapse-free period of 24 weeks. Notably, the relapse rate was 25% in the group treated with rifampin and clarithromycin. Moreover, the authors evaluated the dose-ranging action of telacebec alone and in combination with rifampicin and discovered that rifampicin had no effect on telacebec activity [60]. A different promising molecule is TB47, which, in combination with oral antibiotics (rifampicin, clarithromycin, and clofazimine), can lead to the cure of BU in less than 2 weeks, provided that the treatment is administered daily and in 3 weeks if it is administered twice a week [61].

Fukano et al. assessed the effectiveness of a rifamycin derivative called rifalazil (RLZ) in treating advanced *M. ulcerans* infections using female BALB/c mice. The mice were initially infected with *M. ulcerans* and then administered RLZ orally at various doses. The untreated mice experienced a worsening of symptoms and reached the end-point within 5–8 weeks after infection. Conversely, the mice treated with RLZ demonstrated either an improvement or complete healing of footpad erythema, swelling, and erosion. Within 3 weeks of treatment, the bacterial counts in the treated mice significantly decreased compared to the untreated group. All treated mice survived without any signs of *M. ulcerans* infection. These results suggest that RLZ effectively treats advanced *M. ulcerans* infections in the mouse model [62]. Recently, Pidot et al. investigated the effect of SPR719, the active component of SPR720, a novel aminobenzimidazole, on *M. ulcerans*, *M. marinum*, and *M. chimaera*. SPR719 acts as an inhibitor of the ATPase activity of the DNA gyrase in mycobacteria. The study demonstrated that SPR719 inhibits the growth of the three non-tuberculous mycobacteria, with a minimum inhibitory concentration range of 0.125–4 μg/mL [63].

**Table 2 pathogens-12-01088-t002:** Novel promising antimicrobial drugs for BU.

Antimicrobial Drug	Conclusion
Q203 (telacebec)	a compound that acts on respiratory cytochrome bc1:aa3a single dose of Q203 effectively eliminates *M. ulcerans* in a mouse model of BU, with no recurrence observed for up to 19 weeks after treatment [57,58]
TB47	its mechanism of action is unknownin combination with oral antibiotics (rifampicin, clarithromycin, and clofazimine), can lead to the cure of BU in less than 2 weeks provided that the treatment is administered daily and in 3 weeks if it is administered twice a week [61]
Rifalazil	blocks off the β-subunit in RNA polymeraseeffectively treats advanced *M. ulcerans* infections in the mouse model [62]
SPR719	acts as an inhibitor of the ATPase activity of the DNA gyrase in mycobacteriainhibits the growth of *M. ulcerans* [63]

In vitro studies have indicated that the activity of rifampicin and clarithromycin is increased by beta-lactams [64]. In light of this discovery, a group of researchers have recently proposed a multicenter randomized controlled trial in Benin to compare the standard treatment (rifampicin and clarithromycin, for 8 weeks) with the standard treatment in conjunction with amoxicillin/clavulanate, for 4 weeks. The study began in December 2021 and is to take place over a period of two years. The proposed treatment has the advantage that all antibiotics are administered orally and for a shorter period of time, which can significantly increase treatment adherence and may improve the healing process. Additionally, the required hospitalization days can be reduced, leading to lower costs [42]. A separate study confirms the synergistic action between beta-lactams and rifampicin or clarithromycin. The combination of amoxicillin and clavulanate has quick bactericidal activity and can efficiently eradicate extracellular bacteria, resulting in a decrease in the initial bacterial load and the local levels of the mycolactone toxin. This helps the recovery of the host’s immune response and the clearance of any remaining bacteria in the affected area [64].

Since no topical medication is currently available, a group of researchers looked into the plasma membrane fluidizer, diethyl azelate (DEA), as a possible topical drug. They have observed that DEA inhibits the immunosuppressive activity of *M. ulcerans* and slows down the appearance of ulcers and new lesions while promoting the healing process. *M. ulcerans* has an immunosuppressive effect that is mediated by mycolactone and appears to be inhibited by DEA [65]. In a recent article, poly(hydroxybutyrate-co-hydroxyvalerate) (PHBV) microparticles and gellan gum (GG) hydrogel were utilized to incorporate rifampicin and streptomycin for the cutaneous administration of antibiotics in BU. The obtained hydrogel exhibited a porous microstructure that has an extraordinary ability to retain water (superior to 2000%) and a controlled release of both antibiotics. These results can be the basis of future in vivo studies that will lead to the implementation of a topical treatment for BU and a decrease in adverse effects resulting from the systemic administration of antibiotics [66].

There is no effective vaccine for BU; however, numerous studies have been conducted in recent years with promising results. The initial investigations were carried out by Fenner in the 1950s [67]. The study by Pittet et al. is the first to explore how BCG immunization in humans affects the immune system’s response to *M. ulcerans*. The findings indicate that BCG vaccines generate an immune response to *M. ulcerans* that is similar in quality to the response seen in the case of *M. tuberculosis*. As BU cases may be increasing worldwide, even in countries where BCG immunization is not standard for children, BCG immunization could potentially serve as a valuable preventative measure [68]. However, it should be pointed out that vaccination in endemic areas for *M. ulcerans* have indicated only short-term protection, varying between 6 and 12 months. Looking back at previous studies, it is suggested that there is some level of protection against advanced forms of BU, but the results are not consistent. Various approaches have been employed in mouse studies, yet only a few vaccine candidates have shown better protection than BCG. Recombinant live whole-cell vaccines producing immunogenic antigens present encouraging evidence of protection, although achieving sterilizing immunity has not been proven yet. Additionally, targeting the mycolactone synthesis pathway has demonstrated effectiveness in mouse experiments and may be worth exploring in combination with BCG vaccination [69].

Foulon et al. draws attention to the fact that the diet could represent an adjuvant in the treatment of BU. Recent studies have shown that ketogenic diets help the tissue repair process, thus suggesting that such diets could prove useful for BU patients. They have observed that β-hydroxybutyrate, the main ketone body resulting from the ketogenic diet, inhibits the formation of mycolactone, one of the essential virulence factors in BU. Moreover, this diet promotes the host’s immune response [70]. Ugai et al. have analyzed the influence of nutritional status on the healing of BU. The study included a small number of patients (*n* = 11). The average follow-up period was 19 weeks, and they noticed that patients who have an adequate caloric intake have a faster healing process. It should be taken into account that the human body requires more calories to heal wounds. A total of 60% of patients with an adequate caloric intake achieved wound healing during the follow-up period, compared to only 17% in the case of the group with a low caloric intake. The authors suggest that the correct management of BU should also include educating patients on the principles of correct nutrition [71].

## 6. Conclusions

BU remains a challenging disease; however, if diagnosed early and treated appropriately, the majority of patients can make a full recovery and resume their normal activities. Continued investment in research is critical in order to develop novel therapies and improve BU management. Potential treatments for BU include new antibiotic combinations, immunotherapy, and new topical or systemic drugs. Collaboration among health officials, researchers, and patients is required for the successful control of this skin condition.

## Figures and Tables

**Table 1 pathogens-12-01088-t001:** WHO classification categories [17].

Category I	a single, small lesion < 5 cm in diameter (nodules, papules, plaques, and ulcers)
Category II	single lesions between 5 and 15 cm in diameter, plaque and edematous forms
Category III	single lesions > 15 cm in diameter, multiple lesions, lesions at critical sites (e.g., genital organs or the head and neck), and osteomyelitis

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
