# Peer review of "Recent Advances in the Management Strategies for Buruli Ulcers"

_pathogens, 2023, doi:10.3390/pathogens12091088_

Round 1

Reviewer 1 Report

Popa et al. have reviewed advances in Buruli ulcer management. While it is worthwhile for such advances to be highlighted for the microbiological and clinical communities, the manuscript needs considerable clarification to avoid misunderstandings about recent research as well as aspects of BU epidemiology and pathology.

Abstract: BU does mainly affect people in rural areas of Africa but in Australia, it affects people in suburban communities and in coastal vacation homes rather than rural areas.

Line 14, please remove “the spread of infection and avoiding” and insert “further tissue damage and”. Otherwise, the reader could interpret the statement to suggest that there is person-to-person spread which the authors correctly state elsewhere is not known to occur.

Introduction: Rather than southern Australia, the authors might more precisely state southeastern Australia. Albert Cook was the first to observe the disease (as found in his notes in a hospital library in Kampala) but he never reported the disease (i.e., published his observations).

2. Transmission route: line 51 and 52. Citing a primary reference would be good here.

Line 54: delete infection

Line 67: replace “is” with “may be”

While climate may have a role in the decline in case reports, a possibly more likely reason is failure to detect cases. Ref. 22 (Ahorlu et al.) showed that more cases can be found with active case detection. (This has also been reported to be true for leprosy).

3. Current BU treatment. I am not sure the sentence on lines 77 and 78 is true. A current reference should be added or the sentence should be deleted. The following sentence should also be deleted. Surgery can remove damaged tissue but it cannot really lower the bacterial load. That is why we use antibiotics. The authors should discuss tissue damage that results from paradoxical reactions.

Lines 86-88. The RS treatment would have been for no more than 8 weeks. Follow up might go on for up to 48 weeks. Please re-read reference 13.

Line 95: moxifloxacin is sometimes used

Line 99: again, the treatment is given for no more than 8 weeks and patients are observed for up to a year.

Lines 103-04: Please rewrite these sentences. Patients with BU are either treated with clarithromycin or streptomycin (comparable to amikacin) plus rifampicin. Not sure that amikacin would be of much benefit in dealing with streptomycin resistance.

Line 105: delete initial. According to ref 20 and other papers by the author, the RS and RC combinations are effective in treating all types of BU lesions with smaller ones healing more rapidly than larger ones.

Line 107: treatment should be started early or promptly not quickly

Line 110: Etuaful et al. (2005), for example, has pointed out that many patients are culture negative after less than 8 weeks of treatment. The 8-week recommendation is a cautious or conservative one.

Line 141: Involving rather than Educating

Line 152, Yet, M. tuberculosis (not M. ulcerans here). The bd oxidase in Mtb is functional resulting in an active, if less efficient, terminal oxidase. Most M. ulcerans strains have a mutation in one of the genes encoding the bd oxidase thus making it highly vulnerable to telacebec. Regarding Q203, please also discuss PMIDs: 3106687, 34460302, and 32631818.

Line 160: they proposed

Line 172: indicated rather than proved.

Line 177: and was to take place

In this paragraph, the authors might discuss the rationale of going from a 2-drug regimen that can successfully treat an infection acquired by environmental exposure to a 4-drug regimen – especially when there may be a highly effective monotherapy (telacebec) due to the mutation in the cydA gene.

Line 207: mycolactone is an essential virulence, not inflammatory, factor in BU

Please see sentences that need clarification above.

Author Response

We would like to thank the editors and the reviewers for the careful and thorough review of our manuscript. We have taken into consideration all the suggestions. We consider all the suggestions very useful.

Popa et al. have reviewed advances in Buruli ulcer management. While it is worthwhile for such advances to be highlighted for the microbiological and clinical communities, the manuscript needs considerable clarification to avoid misunderstandings about recent research as well as aspects of BU epidemiology and pathology.

We have carefully addressed all the corrections and suggestions you provided, which has undoubtedly enhanced the overall clarity and coherence of the content. Your expertise and attention to detail have been truly invaluable to us, and we am grateful for the time and effort you invested in reviewing our work.

Abstract: BU does mainly affect people in rural areas of Africa but in Australia, it affects people in suburban communities and in coastal vacation homes rather than rural areas.

Thank you for your valuable comment. We have corrected it.

“Buruli ulcer (BU) is a bacterial skin infection that is caused by Mycobacterium ulcerans and mainly affects people who reside in rural areas of Africa and in suburban communities in Australia”.

Line 14, please remove “the spread of infection and avoiding” and insert “further tissue damage and”. Otherwise, the reader could interpret the statement to suggest that there is person-to-person spread which the authors correctly state elsewhere is not known to occur.

Thank you for your careful review. We have corrected it.

„Early detection and immediate treatment are crucial to preventing further tissue damage and any potential complications, although it is worth noting that access to proper therapeutic resources can be limited in certain areas”.

Introduction: Rather than southern Australia, the authors might more precisely state southeastern Australia.

Thank you for pointing it out. We have corrected it.

Albert Cook was the first to observe the disease (as found in his notes in a hospital library in Kampala) but he never reported the disease (i.e., published his observations).

Thank you for your careful review. We have corrected it.

“Albert Cook, an English physician, was the first to observe the disease , during the late nineteenth century, (as found in his notes in a hospital library in Kampala, Uganda) but he never reported the disease.”

  1. Transmission route: line 51 and 52. Citing a primary reference would be good here.

Thank you for your useful suggestion. We have added a primary reference.

Line 54: delete infection

Thank you for pointing it out. We have corrected it.

Line 67: replace “is” with “may be”

Thank you for pointing it out. We have corrected it.

While climate may have a role in the decline in case reports, a possibly more likely reason is failure to detect cases. Ref. 22 (Ahorlu et al.) showed that more cases can be found with active case detection. (This has also been reported to be true for leprosy).

Thank you for your valuable comments. We have explained that in the text.

“However, the most important factor that influences the incidence of BU is failure to detect cases. Ahorlu et al. showed that more cases can be found with active case detection.”

  1. Current BU treatment. I am not sure the sentence on lines 77 and 78 is true. A current reference should be added or the sentence should be deleted.

Thank you for your careful review. We have corrected the sentence and added a current reference.

“WHO recommends to decide whether surgery is needed 4 weeks after starting the antibiotic treatment”.

The following sentence should also be deleted. Surgery can remove damaged tissue but it cannot really lower the bacterial load. That is why we use antibiotics.

We have deleted the sentence.

The authors should discuss tissue damage that results from paradoxical reactions.

Thank you for your useful suggestion. We have added a paragraph regarding paradoxical reactions.

Lines 86-88. The RS treatment would have been for no more than 8 weeks. Follow up might go on for up to 48 weeks. Please re-read reference 13.

We have re-read the article. It is a systematic review that included various studies in which the treatment duration varied between 8-48 weeks, depending on disease severity; the mean duration of treatment was 8 weeks according to the WHO recommendations.

We have re-written the sentence

“A systematic review examined the efficacy of rifampicin and streptomycin-based therapy for a period of 8-48 weeks depending on disease severity (the mean duration - 8 weeks) and found a 50% cure rate.”

Line 95: moxifloxacin is sometimes used

Thank you for pointing out. We have added “sometimes”.

Line 99: again, the treatment is given for no more than 8 weeks and patients are observed for up to a year.

Thank you for your useful comment. We have corrected it.

Lines 103-04: Please rewrite these sentences. Patients with BU are either treated with clarithromycin or streptomycin (comparable to amikacin) plus rifampicin. Not sure that amikacin would be of much benefit in dealing with streptomycin resistance.

Thank you for your careful review. We have deleted those sentences.

Line 105: delete initial. According to ref 20 and other papers by the author, the RS and RC combinations are effective in treating all types of BU lesions with smaller ones healing more rapidly than larger ones.

Thank you for pointing out. We have corrected it.

Line 107: treatment should be started early or promptly not quickly

Thank you for your comment. We have corrected it.

Line 110: Etuaful et al. (2005), for example, has pointed out that many patients are culture negative after less than 8 weeks of treatment. The 8-week recommendation is a cautious or conservative one.

Thank you for your careful review. We have added the information in the text as suggested.

Line 141: Involving rather than Educating

Thank you for your comment. We have corrected it.

Line 152, Yet, M. tuberculosis (not M. ulcerans here). The bd oxidase in Mtb is functional resulting in an active, if less efficient, terminal oxidase. Most M. ulcerans strains have a mutation in one of the genes encoding the bd oxidase thus making it highly vulnerable to telacebec. Regarding Q203, please also discuss PMIDs: 3106687, 34460302, and 32631818.

Thank you for your useful comments. We have corrected it. We have discussed about Q203, a promising drug in the treatment of Buruli ulcer using the references indicated.

Line 160: they proposed

Thank you for pointing out. We have corrected it.

Line 172: indicated rather than proved.

Thank you for your comment. We have corrected it.

Line 177: and was to take place

Thank you for pointing out. We have corrected it.

In this paragraph, the authors might discuss the rationale of going from a 2-drug regimen that can successfully treat an infection acquired by environmental exposure to a 4-drug regimen – especially when there may be a highly effective monotherapy (telacebec) due to the mutation in the cydA gene.

Thank you for your useful suggestion. We have added the arguments for a 4 drug regimen, but for a shorter period of time as they are presented by Johnson et al.

“The proposed treatment has the advantage that all antibiotics are administered orally and for a shorter period of time, which can significantly increase treatment adherence and may improve the healing process. Additionally, the required hospitalization days can be reduced, leading to lower costs.”

Line 207: mycolactone is an essential virulence, not inflammatory, factor in BU

Thank you for your comment. We have corrected it.

Thank you once again for your valuable contribution to the improvement of our manuscript. We are genuinely thankful for your support in this process.

Gabriela Loredana Popa, and the other authors

Reviewer 2 Report

This review does not follow a formal methodology (i.e., it is not a systematic review) but it is rather an occasional narrative review, addressing prevalence, pathogenesis, transmission, diagnostic testing, and current and future (antimicrobial) therapy.

I see that this paper has improved following imput by reviewers on an earlier version of the paper.

It is remarkable that authors chose te refer to comments but not to original studies that specifically comment on those studies (e.g., when they mention stigma related to BU; and the role of surgery, where they do not mention a randomized study by Wadagni et al, but rather a comment by Johnson; and they miss some of the classical papers on paradoxical reaction;s and societal impact, e.g., on school drop-out, and loss of jobs associated with BU;  ototoxicity associated with streptomycin use by BU patients has been studied by Klis et al; Portaels et al have reviewed studies on animal reservoirs of BU in Africa, back in 2001).

For readers highly familiar with this disease, the review of novel therapies provides a nice update.

I agree that human-to-human transmission is unusual (line 68), but I am aware of at least two reports on human-to-human transmission - one by Exner & Lemperle, 1987;  and one by DeBacker (Lancet 2002; Clin Infect Dis 2003).

In the revision, micro-organisms are not written in italics.

Author Response

We would like to thank the editors and the reviewers for the careful and thorough review of our manuscript. We have taken into consideration all the suggestions. We consider all the suggestions  very  useful.

This review does not follow a formal methodology (i.e., it is not a systematic review) but it is rather an occasional narrative review, addressing prevalence, pathogenesis, transmission, diagnostic testing, and current and future (antimicrobial) therapy.

Thank you for your time and expertise to this review process. We are genuinely grateful for your support in improving our work. Your suggestions/comments greatly contributed to the overall quality of the manuscript.

Thank you for your comment. We have mentioned that it is a narrative review in the text.

I see that this paper has improved following imput by reviewers on an earlier version of the paper.

It is remarkable that authors chose te refer to comments but not to original studies that specifically comment on those studies (e.g., when they mention stigma related to BU; and the role of surgery, where they do not mention a randomized study by Wadagni et al, but rather a comment by Johnson; and they miss some of the classical papers on paradoxical reaction;s and societal impact, e.g., on school drop-out, and loss of jobs associated with BU;  ototoxicity associated with streptomycin use by BU patients has been studied by Klis et al; Portaels et al have reviewed studies on animal reservoirs of BU in Africa, back in 2001).

Thank you for your careful review. We have discussed in the text the suggested articles (by Wadagni et al., Klis et al., Portaels et al.) and we have also mentioned some missed papers regarding paradoxical reaction and social impact.

For readers highly familiar with this disease, the review of novel therapies provides a nice update.

I agree that human-to-human transmission is unusual (line 68), but I am aware of at least two reports on human-to-human transmission - one by Exner & Lemperle, 1987;  and one by DeBacker (Lancet 2002; Clin Infect Dis 2003).

Thank you for your useful comment. We have mentioned in the text the human-to-human transmission.

In the revision, micro-organisms are not written in italics.

Thank you for pointing it out. We have revised it.

Thank you once again for your valuable contribution to the improvement of our manuscript. We are genuinely thankful for your support in this process.

Gabriela Loredana Popa, and the other authors

Round 2

Reviewer 1 Report

I would suggest better phrasing on line 11 - I think it would be most accurate if the authors could insert “and beach resort” after suburban. The suburban areas are rather new in the epidemiology of Buruli ulcer in Australia but beach resort communities are a long-standing and continuing area of risk.

For the full review report, please see attached file. 

Please see sentences that need clarification above.

Author Response

We would like to thank the editors and the reviewers for the careful and thorough review of our manuscript. We have taken into consideration the suggestions. We consider all the suggestions very useful.

I would suggest better phrasing on line 11 - I think it would be most accurate if the authors could insert “and beach resort” after suburban. The suburban areas are rather new in the epidemiology of Buruli ulcer in Australia but beach resort communities are a long-standing and continuing area of risk.

We thank the Reviewer for this insightful comment. We have adapted the text to match the reviewer’s observation and have added a recent relevant citation. Lines 11-12, lines 45-46.

The following Reviewer’s

line 12: add typically – not all cases present as a papule or nodule

line 33: change some of the to such as

line 41: modify: when the healing process begins, after effective antibiotic treatment,

line 83: unusual human-to-human transmission after an accidental injection or a bite.

Line 86: cause

Line 105 may be

Line 248: were not

Line 254: emphasized

Line 345: appearance rather than apparition

We thank the Reviewer for the careful reading of the manuscript and entire support (for a a better manuscript).

Line 149: fluorescent TLC has been reported in mouse studies, see PMID: 24392174. It is true that it is not easily done in all labs and further studies are necessary.

We thank the reviewer for their comment. The suggested study was also cited, and the text was adapted.

Line 207: Klis et al., using audiometry rather than patient report,

Highlighting the methodology of the study was a very good suggestion of the Reviewer.

Line 285: revealed that, due a mutation in the genes encoding the bd oxidase, these strains lacked an alternate terminal oxidase, rendering these predominant M. ulcerans strains highly vulnerable to Q203.

This is an excellent contribution that the Reviewer has made. This is indeed a difficult to grasp concept and it is now better explained in the text.

All the italicization corrections were made as per the Reviewer’s kind suggestions.

line 28: italicize M. marinum

Line 318: italicize M. marinum and M. chimaera

Thank you once again for your valuable contribution to the improvement of our manuscript. We are genuinely thankful for your support in this process.

Kind regards,

Gabriela Loredana Popa, and the other authors

Reviewer 2 Report

the authors made a fair attempt to address questions and queries. Although I still see some unbalance in citations, I feel the authors cover the topic fairly well in this revised version.

no problem with English language

Author Response

We would like to thank the editors and the reviewers for the careful and thorough review of our manuscript. We have taken into consideration the suggestions. We consider all the suggestions very useful.

the authors made a fair attempt to address questions and queries. Although I still see some unbalance in citations, I feel the authors cover the topic fairly well in this revised version.

We thank the reviewer for judging our review on the subject as fair.

As per the instructions of the Reviewer we have added citations 7 and 32, which we hope the Reviewer finds suitable for describing the epidemiology and alternate diagnostic methods.

We would like to also kindly thank the Reviewer for the thoughtful reading of the manuscript.

Thank you once again for your valuable contribution to the improvement of our manuscript. We are genuinely thankful for your support in this process.

Kind regards,

Gabriela Loredana Popa, and the other authors
